

# Modelling water quantity and quality for integrated water cycle management with the WSIMOD software

Barnaby Dobson[1], Leyang Liu[1], Ana Mijic[1]

[1]Department of Civil and Environmental Engineering, Imperial College London, UK

*Correspondence to*: Barnaby Dobson (b.dobson@imperial.ac.uk)

**Abstract**

Problems of water system integration occur when a model's boundaries are too narrow to capture interactions and feedbacks across the water cycle. We propose that integrated water systems models are required to overcome them, and are necessary to understand emergent system behaviour, to expand model boundaries, to evaluate interventions, and to ensure simulations reflect stakeholder goals. We present the Water Systems Integrated Modelling Framework (WSIMOD) software as one such approach and describe its theoretical basis, covering the node and arc nature of simulations, the integration framework that enables communication between model elements, and the model orchestration to customise interactions. We highlight data requirements for creating such a model and the potential for future development and refinement. WSIMOD offers a flexible and powerful approach to represent water systems, and we hope it will encourage further research and application into using model integration towards achieving sustainable and resilient water management.

**Plain Language Summary**

Water management is challenging when models don't capture the entire water cycle. We propose using integrated models facilitates management and improves understanding. Thus, we introduce a software tool designed for this task. We discuss its foundation, how it simulates water system components and their interactions, and its customization. We provide a flexible way to represent water systems, and we hope it will inspire more research and practical applications for sustainable water management.

## 1 Introduction

Water fluxes and their pollution concentration are influenced by the interactions and management of all components that make up the human-altered water cycle, including but not limited to upstream rivers, reservoirs, freshwater treatment, distribution networks, residential and non-residential water consumers, foul sewers, urban drainage, storm sewers, wastewater treatment works, groundwater, agriculture, and hydrological catchments (and the many physical and operational processes within each component). The importance of this interconnectedness is most evident in rivers, which reflect the overall condition of the catchment system since they aggregate behaviour over such large areas (Dobson and Mijic 2020; Kirchner 2009). River catchments are rarely dominated by the behaviour of any specific component or any individual stakeholder's decisions. For example, 60% of English catchments that do not achieve a 'good' status in the Water Framework Directive (WFD) do so because



of multiple different pollution sources, with wastewater infrastructure and agriculture being the most prevalent
drivers, each affecting 50% of catchments. (Environment Agency 2020b). The implication is that a modelled
representation focussing on any individual component is unlikely to give accurate estimates of impacts beyond
that component (Beven 2007; Blair et al. 2019; Dobson, Wagener, and Pianosi 2019; Schmitt and Huber 2006).
Furthermore, estimates within a subsystem representation may be inaccurate if sensible boundary conditions
cannot be defined, something that water managers are highly sensitive to (Höllermann and Evers 2017). We term
these problems of water systems integration, and it follows that understanding the water cycle as a whole is needed
to address them. Models that take such an approach will better capture component boundaries and the wider
impacts of stakeholder decisions, ultimately enabling more accurate representations of water quality in rivers,
which is essential to effectively manage, for example, water supply (Mortazavi-Naeini et al. 2019) and
biodiversity (Dobson, Barry, et al. 2022).
In this paper we introduce the theoretical underpinning behind a novel method for modelling integrated water
systems to address these challenges. Firstly, the need for integrated water cycle simulation models is explained,
including their current coverage. The importance of parsimonious representations within an integrated model is
then discussed, along with the methods used to achieve integration.
The environmental modelling research community has responded to problems of water systems integration
primarily through computer simulation models (Bach et al. 2014; Best et al. 2011; Douglas-Mankin, Srinivasan,
and Arnold 2010; Rauch et al. 2017; Tscheikner-Gratl et al. 2019; Whitehead, Wilson, and Butterfield 1998). We
distinguish an integrated water system modelling approach from a system dynamics approach (see Zomorodian
et al., (2018)) by further specifying that component representations must have a physical basis, which is needed
to link observational data to model behaviour and to capture interventions (e.g., new infrastructure or changes to
operations). We define integrated water system models as those which link component representations to capture
and understand the complex interactions and feedbacks that occur between components. We categorise the four
key goals of these models: (1) to understand which fundamental processes drive emergent behaviour at a whole-
water system scale; (2) to avoid simulation inaccuracies caused by narrow boundary conditions; (3) to test
interventions to the physical system or operational behaviour in order to understand their water cycle wide impacts
or interactions and (4) to capture impacts that align more closely with desired water system outcomes, in addition
to performance indicators of individual components. For example, in-river pollutant concentration is a better
indicator of wastewater system performance than the more typically monitored number of sewer spills (Giakoumis
and Voulvoulis 2023).
In addressing problems of water systems integration, existing modelling approaches have made significant
progress. Bach et al. (2014) set out a comprehensive typology for integrated urban water systems modelling.
However, among the reviewed models, only CityDrain3 (Burger et al. 2016), WEST (Vanhooren et al. 2003), and
SIMBA (IFAK 2007) can represent receiving water bodies (i.e., rivers), which is where the importance of an
integrated representation is most pronounced. Furthermore, due to the urban focus of these models, the ability to
simulate pollution concentrations in receiving waters impacted by upstream catchments is highly limited yet is
central to quantifying in-river impacts (Liu, Dobson, and Mijic 2022). A more recent effort to characterize
integrated water systems modelling places importance on in-river conditions (Tscheikner-Gratl et al. 2019) and
present a comprehensive review of urban and rural water cycles and their impacts on rivers. However, the
reviewed modelling approaches omit some key factors: the importance of water resources infrastructure, which



play a significant role in concentrating pollution during low flows if abstractions take place; the relevance of
groundwater, which provides baseflow to dilute pollution during critical low flow periods; and consideration of
agricultural processes and associated pollution that results from them, which is a critical source of water pollution
worldwide (Mateo-Sagasta et al. 2017; Tang et al. 2021), and the second most common catchment pollution source
in England (Environment Agency 2020b). Integrated models that capture groundwater and agricultural processes
are present in the modelling literature, such as INCA (Whitehead et al. 1998) and HYPE (Lindström et al. 2010),
however, in contrast, these are limited by their ability to capture urban systems. Thus, while water systems
integration is well-served from a rural or urban modelling perspective, we identify that there is not yet an approach
that offers a self-contained representation to capture all key processes required to model in-river water quality at
a whole-water cycle scale.
A further critical factor in creating an integrated water systems model is how components are represented. In
general, current approaches have favoured identifying pre-existing detailed component representations which are
then integrated (Schmitt and Huber 2006). For example, DAnCE4Water (Rauch et al. 2017), which is the most
comprehensive application to date, includes high resolution and sophisticated models for a wide range of urban
components. However, as more and more components are captured by integrated modelling, it becomes
increasingly difficult parameterising such detailed models. Simply combining separately calibrated models
provides no guarantee of performance as a whole (Lee 1973). Meanwhile, integrated models typically have many
parameters that may compensate each other, thus making calibration a challenging and risky process (Voinov and
Shugart 2013). An alternative approach is to forego calibration altogether by adopting parsimonious models with
fewer parameters and ideally deriving those parameters from best available data (Dobson et al. 2021). Although
complicated modelling approaches are needed for tasks such as design, these approaches are also more difficult
to apply widely and thus may hinder the goals of integrated water systems modelling. For example, building
scientific understanding requires repeated testing of an approach in various locations, and customising the model
to match local conditions is essential when representing interventions. Therefore, a modelling approach that can
be easily deployed on a wider scale is of significant benefit to problems of water systems integration.
To ensure that a range of water system configurations can be accommodated, flexibility or customisability must
be incorporated into the approach for integration. Integration approaches vary broadly between tightly coupled
and loosely coupled. In a tightly coupled approach equations and interactions are pre-defined to create a self-
contained integrated representation, such as with JULES (Best et al. 2011) or INCA (Whitehead et al. 1998).
While, in a loosely coupled approach, component representations are self-contained, and the integration occurs
by facilitating their interactions, filling the role as a message passing interface, such as with OpenMI (Harpham,
Hughes, and Moore 2019) and DAnCE4Water (Rauch et al. 2017). Belete et al. (2017) describe the arrangement
of components and their interactions as integrated model orchestration, highlighting that different orchestrations
are suitable for different applications. While looser coupling provides greater control over orchestration, and thus
greater ability to customise and capture a wide variety of systems, it also creates a higher user burden to set up
and understand many subsystems, considered to be a key barrier to the uptake of such approaches (Zomorodian
et al. 2018). Conversely, a tightly coupled model that represents the same components as a loosely coupled one
may be easier to set up but typically offers less control over orchestration. In the middle ground is an integrated
representation that gives flexibility around orchestration but comes with self-contained components that do not
need to be onerously setup by a user, such as the CityDrain3 software for modelling urban drainage systems



(Burger et al. 2016). We propose that this middle ground is the most beneficial for a modeller and believe that
such an approach to integration is the most productive avenue towards creating highly flexible, user-friendly
models of the integrated water cycle.
The concepts introduced above suggest that, for many problems of water systems integration, capturing a broad
representation of the water cycle and interactions between its components is equally important as detailed
component representations. We have created a tool to implement this modelling philosophy, the Water Systems
Integrated Modelling framework (WSIMOD), which is an open-source Python package for flexible and
customisable simulations of the water cycle that treats the physical components of the water cycle as nodes
connected by arcs that convey water and pollutant flux between them. The software source code and online
tutorials are published by Dobson, Liu, and Mijic (2023a), in contrast, this paper presents WSIMOD's theoretical
underpinning with a discussion on model setup and of integrated water system modelling in general. To address
the difficulties in application associated with integrated modelling mentioned above, WSIMOD contains a library
of built-in component representations covering a more complete water cycle coverage than any identified
integrated models, and a default but customisable orchestration adjudged to be suitable for many catchments and
regional water systems coordination. Where possible these representations are based on parsimonious and peer-
reviewed models. Extensive model documentation with worked examples is provided online (Dobson, Liu, and
Mijic 2023b), enabling users to gain confidence and become familiar with using WSIMOD.
**2    WSIMOD**
WSIMOD is an integrated modelling framework that provides ready-to-use objects (nodes, arcs, water stores, and
model orchestration) that are suitable for a wide range of water systems and described in greater detail in the
following sections. However, WSIMOD is not intended to be a one-size-fits-all solution, indeed, the ubiquity of
non-textbook water systems led us to create a more customisable modelling approach in the first place. This paper
describes the theory behind WSIMOD in general and user-friendly terms, avoiding the use of equations and
technical details, while further documentation can be found online (Dobson, Liu, and Mijic 2023c). The WSIMOD
framework is implemented in Python 3, which is widely practiced in the environmental modelling community and
facilitates quick setup and easy customisation. WSIMOD is the combined effort of many studies conducted as
part    of    the    CAMELLIA    (Community    Management    for    a    Liveable    London)    project
(https://www.camelliawater.org/), which are linked to relevant sections of the model description to highlight the
range of possible applications.
An example WSIMOD model is shown in Figure 1, demonstrated for Luton, UK, selected for illustration because
its water cycle is reasonably self-contained. WSIMOD uses object-oriented programming (OOP), which classifies
components by common attributes and behaviours (classes), thus facilitating customisation or the introduction of
new behaviours. All objects in WSIMOD are a subclass of WSIObj, which predefines efficient arithmetic
operations for water quality and volume, however users will typically instead interact with the subclasses
described in the following sections. Additionally, users may customise a model's high-level control over how
interactions take place within a timestep, or the model's orchestration, which is a unique feature of WSIMOD
(Belete et al. 2017). Thus, while Figure 1 depicts one possible arrangement and selection of nodes and arcs, a
wide variety of water systems can be represented.



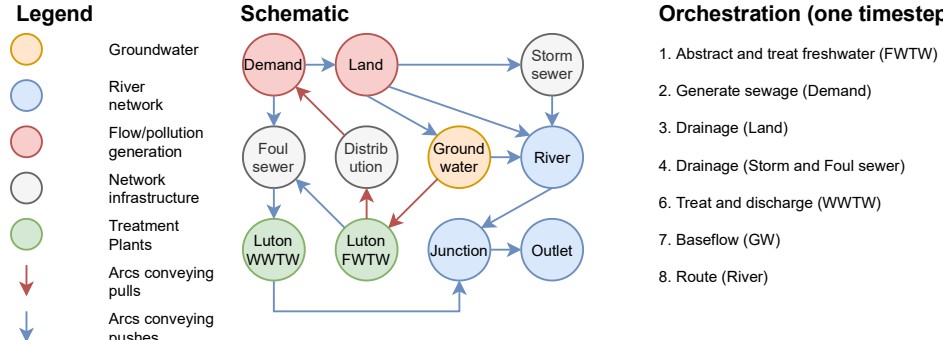

**Figure 1: An example WSIMOD model for Luton, UK. Orchestration is shown demonstrating the high-level functions called for each timestep. Nodes are shown as circles and arcs as arrows. WWTW stands for Wastewater Treatment Works and FWTW stands for Freshwater Treatment Works.**

## 2.1 Nodes represent water cycle components

Physical representations of the different components in the water cycle are typically implemented as WSIMOD nodes, Figure 1. In the software implementation, all nodes are instances of the Node class or its subclasses. Formulation of components as nodes using OOP draws heavily on the CityDrain3 software (Burger et al. 2016). Our generic definition allows nodes to represent diverse entities, for example, a collection of manholes representing a region of sewer network or individual manholes that can be connected to represent a sewer network; as demonstrated in (Dobson, Watson-Hill, et al. 2022). In this section we describe the Node class, summarise the existing node subclasses currently implemented in WSIMOD, and describe how to customise them.

### 2.1.1 The Node class

The Node class in WSIMOD is a generic class that is expected to be the parent of any component represented. The class captures three key behaviours. Firstly, it predefines defaults for much of the interaction functionality later described in Section 2.2. Secondly, it contains a variety of useful functions to interact with other nodes via arcs (see Section 2.2.1). Thirdly, it enables mass balance checking (see Section 2.3.2). For these reasons, physical components should generally be implemented in WSIMOD as a subclass or variation of a Node, even if the computational implementation is simply a wrapper for another, pre-existing, model. Although the base Node class does not implement any physical processes, it may serve as a junction for branches or convergences in the water system, such as river bifurcations and confluences.

### 2.1.2 Node subclasses

To ensure WSIMOD is as easy to be implemented as possible, a variety of water cycle components have been developed in Python as Node subclasses. We summarise these components in Figure 2, and recommend viewing the online documentation for full details, which provides a library of documented components as well as tutorials on their use (Dobson et al. 2023c). As anticipated in the introduction, these components are designed for parsimony and can be instantiated with as few parameters as possible, thus minimising data requirements and



maximising utility. We note that this summary is up to date as of time of writing, however the Imperial College
London Water Systems Integration research group will be continually upgrading and adding new functionality to
WSIMOD.

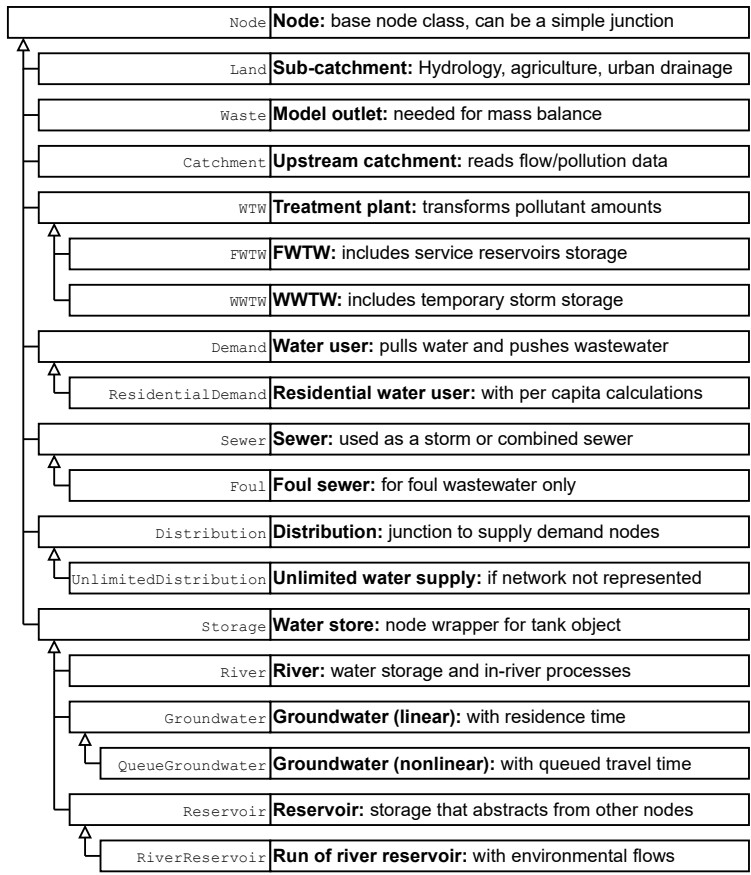


**Figure 2: An inheritance diagram of components implemented as nodes in WSIMOD, arrows indicate that a node is a**
**subclass of another node. The courier text is the name of the node in WSIMOD.**
Many node types include water stores, which are sufficiently prevalent in water systems to warrant their own class
in WSIMOD, referred to as a Tank, with further details provided in A1.1 Tank object to generalise water stores.
It's important to note that a tank is not a node, but a node may have a tank. The number of tanks a node subclass
may have varies depending on what it represents, with some having none (e.g., Demand), one (e.g., Reservoir),
or multiple tanks (e.g., Land). If a user requires a tank that can act as a node, they can utilize the Storage class to
achieve this functionality.
**2.1.3   Customising nodes**
We do not expect that the provided WSIMOD components will be sufficient to cover every water system or be
suitable in cases where detailed representations are necessary. Where possible, users should consider customising





model orchestration (Section 2.3) or component interactions (Section 2.2.3) to represent these cases. However,
new physical subsystems may also be represented based on those defined in Figure 2. The variety of techniques
to customise subclass behaviour in OOP are outside the scope of this paper but have been discussed extensively
elsewhere (Gamma et al. 1995). In general, these techniques aim to avoid duplication of effort, because most
functionality will be predefined in an existing class that can maintain specific interactions between existing
subclasses. We enthusiastically encourage users to contribute to the open-source package to create a better
software for the wider environmental modelling community, though request that contributors our read online
guidelines before doing so to avoid highly duplicative and bloated components that are likely to confuse others.
If a pre-existing model aims to be integrated and it would be too significant a programming effort to reimplement
under this philosophy, then we would request that wrappers are used to treat the model as a node and interface
with the existing software.
To highlight the flexibility offered by WSIMOD, we briefly discuss some examples of complex behaviour being
captured through node customisation. In Dobson et al., (2021), demand nodes were assigned time varying
population, water use behaviour, and pollutant generation to capture changing commuter patterns in London that
resulted from the COVID-19 pandemic. In Liu et al., (2023a), the hydrological processes in land nodes were
customised in a variety of ways to represent nature-based solutions. We note that these examples are provided
with open-source code but are not included in the WSIMOD repository because they have significant data
requirements to set up that are context specific, and thus are unlikely to be generalisable to a wide range of cases.
In the software documentation we also provide a how-to guide for node customisation (Dobson et al. 2023c).
**2.2    Integration framework**
The WSIMOD integration framework facilitates interactions between nodes by serving as a message passing
interface that transfers information relating to water quality and quantity. It was developed in an application to
London's water cycle at a wastewater catchment scale (Dobson et al. 2021). It draws significantly on the OpenMI
(Harpham et al. 2019) and Open MPI (Graham et al. 2006) interfaces but has been tailored to water systems by
providing a variety of built-in behaviours. The integration framework consists of three key concepts: arcs, pushes
and pulls, and requests and checks (demonstrated in Figure 3 and described in the following sections). Arcs are a
class that facilitate interactions between nodes but can also represent physical entities (e.g., pipes). Arcs convey
both water quality and quantity fluxes, which are discretised and packaged together, based on concepts from
CityDrain3 (Burger et al. 2016). Pushes and pulls differentiate between the directionality of an interaction.
Requests and checks differentiate between information passing that simulates the movement of water (requests)
and that which does not (checks). While we do not recommend changing the integration framework itself, we
provide a generic method to accommodate a wider variety of interactions in Section 2.2.3.



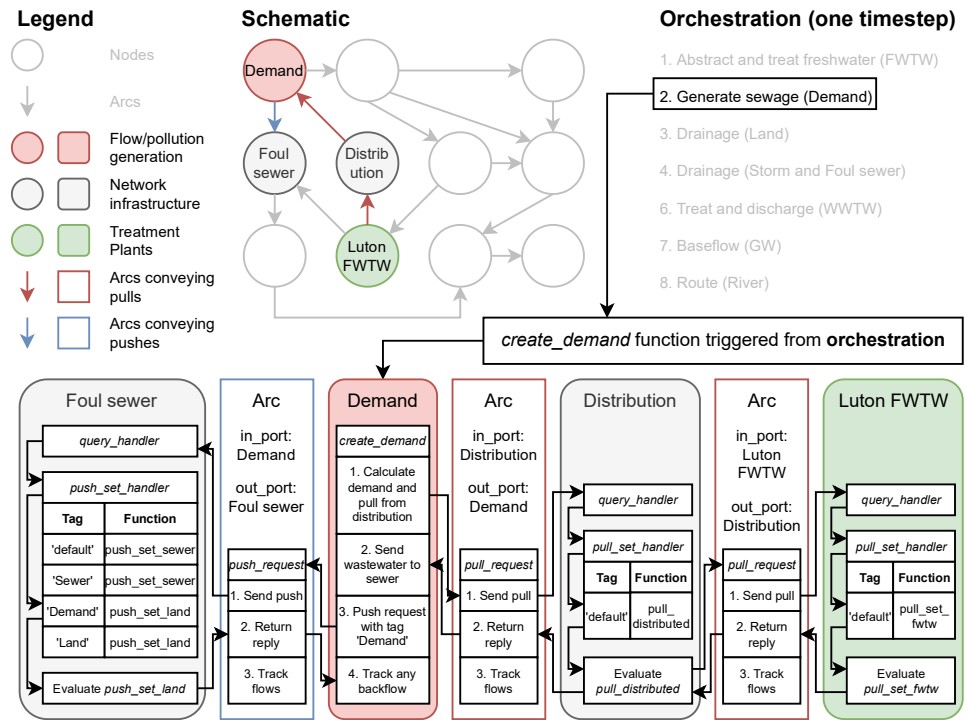

**Figure 3: An example of the WSIMOD integration framework, illustrated through the automatic behaviour triggered when the *create_demand* function is called by a Demand node during orchestration. The node pulls water via an arc from a distribution network and pushes foul wastewater via an arc to a sewer. The figure further illustrates the use of handlers and tags to customise interactions between nodes. Italicized text indicates that it is a function. Nodes are shown as circles or rounded squares, while arcs are shown as coloured arrows or sharp coloured squares.**

### 2.2.1 The Arc class and fluxes

Arcs are a class to establish connections between nodes. They transmit all message passing and track fluxes when requests are made. Arcs can have a capacity property that limits the flow in a given timestep, which can also be customised to be dynamically calculated, for example, to implement Manning's equation along pipes (Dobson, Watson-Hill, et al. 2022). Additionally, if multiple arcs are linked to a single node, they can be assigned a preference attribute. This enables the node to prioritize certain arcs over others. For example, a sewer node may connect to a wastewater treatment plant (WWTW) and to a river via a sewer spill. In this case, the spill arc could be assigned a low preference to ensure that it is only utilized if the WWTW cannot accept any water.

Fluxes in WSIMOD are described in discrete packages called Volume-Quality Information Packages (VQIP). A VQIP is a dictionary that contains entries for volume and all simulated pollutants. Calculations in WSIMOD are typically performed on VQIPs rather than simply flows, thus ensuring simulation of both water quantity and quality. The core parent class (WSIObj) of all WSIMOD classes provides functions to perform basic operations with VQIPs in place of the normal arithmetical operations that would typically be only performed on flows or volumes. VQIPs track water quality as mass rather than as concentration to accommodate cases where pollutants



are being moved with no associated water quantity, except for non-additive variables such as temperature or pH.
WSIMOD refers to anything tracked in a VQIP that is not water volume as a pollutant, however this should be
interpreted as a water quality constituent, and does not imply that everything simulated (e.g., temperature or
dissolved oxygen) is a pollutant from an environmental/management perspective.
WSIMOD can simulate any number of pollutants in a mass balance approach, also referred to as conservatively,
provided their sources into the water cycle can be identified and quantified. However, bio-chemical changes for
pollutant decay can also be represented, see Appendix 1, A1.2 Non-flux pollutant changes. Other more
complicated pollutant transformation can be captured on a case-by-case basis, for example to capture nutrient
cycling in the soil pool, using on equations from Liu et al., (2022).
A further consideration commonly required in water systems is that of travel time of water, which requires its own
specific implementation due to the discrete nature of flux and VQIPs in WSIMOD, further details are provided in
A1.3 Travel time of water.

### 2.2.2     Types of interactions: pushes/pulls and requests/checks

In order for a simulation to occur in a non-tightly coupled integrated representation, something must trigger the
interactions that are conveyed via arcs. High-level controls that govern these behaviours are described as model
orchestration (see Section 2.3), however a key benefit to using this integration framework is that the user does not
have to predefine all possible interactions in advance. Because information transmitted by arcs automatically
triggers further information transmission in connected nodes, a user may customise their nodes, arcs, and
orchestration, without onerously updating every possible interaction that may take place in the model, as
visualised in Figure 3.
To represent a wide variety of behaviours, WSIMOD categorises interactions based on directionality of intent and
whether they represent flux or not. Directionality of intent refers to either cases when a node has water that must
be sent somewhere, called a push, or when a node needs water from somewhere, called a pull. Interactions between
nodes, whether pushes or pulls, may convey flux of water and pollutants, and simulate the movement of water, or
they may convey non-flux information necessary for achieving realistic simulations. An interaction conveying
flux is referred to as a request, while a non-flux interaction is a check.
Pushes occur when a node needs to discharge water. For example, when a wastewater treatment works (WWTW)
must discharge effluent to a river, or a catchment must discharge runoff downstream. In general, push scenarios
are more common in water systems because water travels from upstream to downstream. Meanwhile, pulls occur
when a node requires water. For example, when a farmer abstracts water from a borehole to fill their irrigation
reservoir. Pulls typically represent human-related effort to move water in a non-natural way.
A request occurs when a node intends to push or pull a certain amount of water to or from another node, regardless
of the connected node's current state. For example, in the pull case, a demand node will intend to satisfy its entire
water needs by pulling water from the distribution network, and this intention is independent of the availability of
water in the distribution network. For example, in the push case, a demand node will always intend to send the
entire volume of its foul water to a sewer system, even if the sewer cannot accommodate the full amount. All
water flux in WSIMOD is ultimately simulated by requests, however most cases are not as straightforward as the
above examples, and nodes often require additional information about the state of giving or receiving nodes to
calculate their requests. Interactions passing such non-flux information are referred to as checks.





Generically a check is any kind of non-flux information passing between nodes, it enables a node to use the state
of the nodes that it interacts with to calculate its requests. For example, when a freshwater treatment works
(FWTW) can draw water from multiple viable reservoirs collectively containing more water than needs to be
treated, a calculation is required before requesting water. The FWTW will send checks to the connected reservoirs
to determine their available water capacity and calculate the appropriate ratio to satisfy its treatment demand. Only
then will the FWTW send requests for the required water.
**2.2.3    Default and customised interactions**
During simulation, a node needs to make responses when it receives a push/pull request/check, which we term as
a reply. We formulate four types of predefined replies that are widely observed in node interactions in the water
systems. These default replies allow nodes to interpret the responses of requests/checks sent to the interacting
nodes without the need to understand their detailed behaviour. These defaults are set out in Table 1 with examples.
**Table 1: The default reply from each kind of component interaction with an example.**

| | Push | | Pull | |
|---|---|---|---|---|
| | Reply | Example | Reply | Example |
| Request | Amount not received | A sewer sends a push request to a WWTW. The WWTW calculates the available capacity, updates its state variables to represent the increased throughput, and replies with how much water from the request that could not be received. | Amount sent | A FWTW sends a pull request to a reservoir. The reservoir calculates how much of the request can be met, updates its state variables to decrease the current volume, and replies with the amount of water abstracted. |
| Check | Maximum volume available to push | A sewer sends two push checks to two downstream sewers (required to calculate what proportion to discharge water to them). The two sewers reply with total amount of water that they can each receive. | Maximum volume available to pull | A FWTW sends two pull checks to two reservoirs (required to determine what proportion to pull water from them). The reservoirs reply with each of their current abstractable volumes. |

While these default interactions are likely to accommodate much of the information passing required to simulate
an integrated water cycle, further customisation may be necessary to allow specific nodes to respond differently
to others. For example, a sewer node might respond differently to a push request from another sewer than from a
land node because sewer to sewer travel time may be calculated differently than land runoff to sewer travel time.
Furthermore, the default check behaviour which transmits capacity information may not be sufficient for a
component to calculate where to push/pull water. For example, due to the importance of head in determining flow,
the amount of water that a floodplain can discharge would not be based on the receiving river's capacity but
instead its head (Liu et al. 2023a).



To define what a node does in reply to an interaction, all WSIMOD interactions pass through 'handlers' that are
associated with a 'tag'. A handler is a Python dictionary belonging to a node, and the tag is a key to the handler
that determines which function is called during a given interaction (see Figure 3). All nodes must have handlers
which contain a 'default' tag, thus enabling any node to interact with any other node. However, additional tags
and replies can be added to enable different behaviours based on the type of node that is interacting with it. An
example interaction for a Demand node draining to a foul Sewer node is given in Figure 3. The generation of
wastewater in the Demand node is triggered by orchestration and sent as a push request, via an arc, to a sewer
node, with a tag 'Demand' to indicate how the sewer node should reply. The sewer node has predefined behaviour
for tags 'default', 'Sewer', 'Demand', and 'Land', with different functions associated with these tags. The sewer
node uses the handler to identify that it should evaluate the 'push_set_land' function (which calculates the travel
time through the sewer using the same equation as is used for calculating travel time of runoff arriving from a
land node) and returns the reply via the arc. The base Node predefines simple default handlers that enable it to
convey interactions to connected nodes, for example, a push request to a Node will trigger push requests to
connected nodes, in this way the Node can behave as a simple junction. A how-to guide to explain this behaviour
in greater detail, with examples, is available in the documentation (Dobson et al. 2023c).
**2.3    Orchestration to manage and enable simulation**
As defined by Belete et al. (2017), orchestration is how nodes and their interactions are managed and enabled.
The default steps that we consider to be important to include in orchestration for many water systems are shown
in Figure 1. In integration frameworks such as OpenMI, all interactions in a simulation occur by nodes triggering
pulls to other nodes, orchestrated by a single external pull each timestep (Harpham et al. 2019). Meanwhile, in
WSIMOD, a finer level of control of orchestration is given to a user. We argue that the WSIMOD approach is
better at capturing complex within-timestep behaviours.
In Figure 4, building on the system shown in Figure 1, we provide a motivating example of how it is more efficient
to customise the behaviour of water systems by customising orchestration, rather than customising/adding nodes
or arcs, as would be required in other integration frameworks. We consider a common case for an integrated water
system of requiring the simulation to perform downstream re-abstraction of wastewater effluent, visualised by the
red arc in Figure 4. Wastewater re-abstraction (water from W reaching A) requires available wastewater at W to
be matched against demand for re-abstracted wastewater at A, both calculations involving the interconnected
node, J.



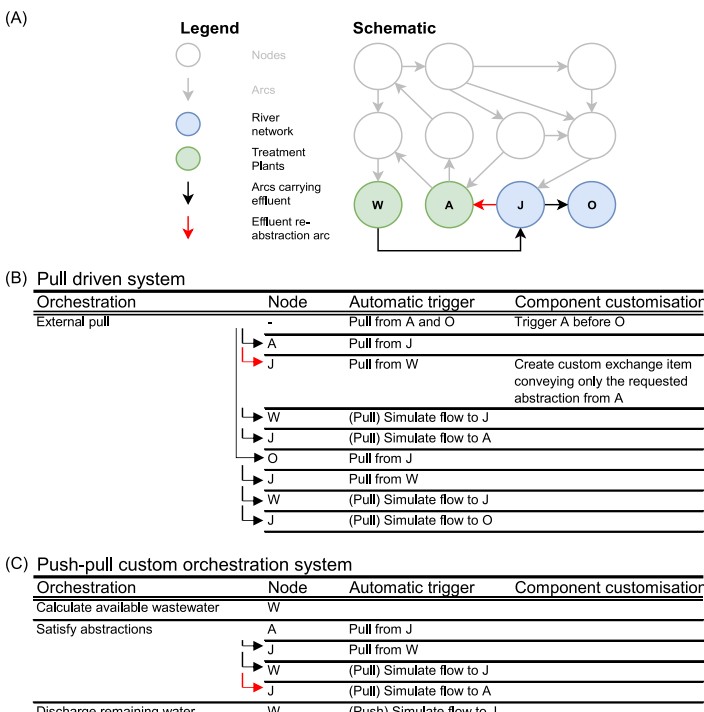

**Figure 4: (A) an example water system based on the Luton example from Figure 1 that includes re-abstraction of treated effluent (red arc). (B) the model steps to achieve re-abstraction of effluent when using a pull-driven integration framework, of the kind used in OpenMI, without customisable orchestration, (C) and when using a push-driven integration framework, of the kind used in WSIMOD, with customisable orchestration.**

To capture such an interaction without customising orchestration and using only pulls, as would be required with an OpenMI approach (Figure 4 (B)), customisation of J would need to specify whether the pull is originating from A or O, and this information would need to be conveyed to W. This is because, if the pull is intended to route the system for that timestep (i.e., it is originating from O via J), then W should release all its effluent, while if it is intending to satisfy the demand (i.e., it is originating from A via J), it should only release enough to meet A's requirements. Additionally, the external pull that initially triggers all interactions must be customised to pull from A before O, so that W does not fully route before abstractions can be made.

Instead, a user with a broad overview of the water cycle may more easily accommodate such behaviour if given high-level control over the model orchestration (Figure 4 (C)). During a timestep the orchestration can specify that W first calculates its treated effluent without discharging it into a receiving river, abstractions are triggered at A, which draws water from W via J, and then W discharges any remaining effluent into its receiving water. This flexibility in orchestration enables representations of a wide variety of water systems while minimizing changes to the behaviour of underlying components.



### 2.3.1    The Model class

A Model class contains all nodes, arcs and forcing data, and provides a default orchestration adjudged to represent a wide variety of water systems. Whether the WSIMOD default orchestration is used or whether an entirely new orchestration is defined, we recommend the use of a Model class for a variety of reasons. Firstly, built-in load and save functionality enables easy sharing and editing of a specific model. Secondly, it contains a run function that can carry out orchestration, perform mass balance checking (see following section), store simulation results, and trigger end of timestep functions. Thirdly, it enables easy collection and grouping of nodes to facilitate orchestration, for example, all WWTWs are referenced by dictionary belonging to the Model class so that they can easily be triggered during a timestep. A tutorial in use of the Model class is provided in the online documentation (Dobson et al. 2023c).

### 2.3.2    Software quality control

Due to the integrated nature of the water systems that WSIMOD simulates, it can be easy to introduce errors that are difficult to spot. We provide extensive unit testing in line with best software development practice, which enables ensuring that changes to code do not introduce unintended behaviour changes. However, a further safeguard against this is a unified method for mass balance error checking. Both nodes and arcs have predefined lists of functions to calculate the total inflows, total outflows and change in any storage. Any newly defined behaviour for nodes and arcs must then also consider how mass balance checks will be impacted and thus update the lists of functions associated with inflows/outflows/change in storage.

Because the core functionality of WSIMOD that performs mass balance checking is indifferent to units (see Section 2.3.3), and because some pollutants exist in far smaller quantities than others, it is possible that both incredibly small and large numbers may be present in VQIPs in the model. Therefore, mass balance checking compares at the magnitude of the largest value of inflows/outflows/storage for a given pollutant or water volume for a given model element in each timestep. Any discrepancy that is larger than a user specified value is reported, although because of floating point accuracy some discrepancies will be unavoidable. We encourage users to exercise common sense to not chase down incredibly small discrepancies while seeking to understand larger discrepancies, which are usually indicative of some implementation error. The user control over orchestration typically makes debugging WSIMOD models easier than most integrated models, as a user may step through each set of triggers in the orchestration and recheck mass balances until the error occurs.

### 2.3.3    Units and timesteps

While the core of WSIMOD is agnostic of units, many nodes are parameterised and assume input data is in SI units, thus we recommend use of SI units throughout. If extensive work would be required to re-implement equations in new units, we recommend converting before and after the calculations from/to SI units.

Because the timestep size will vary depending on the application of WSIMOD, nodes do not necessarily make assumptions about timestep, instead requiring the user to define the timestep that is consistent with their parameters and input data. This enables significant flexibility in representations that can enable studies investigating the impact of timestep size on simulations (Dobson, Watson-Hill, et al. 2022). We note that the




detailed bio-chemical processes used in agricultural surfaces and rivers assume a daily timestep. In addition, these
processes are developed to focus on pollutants associated with crop nutrient cycles.

## 3    Demonstration

Throughout this paper, numerous WSIMOD case studies are referenced. However, to illustrate the scope and
purpose of WSIMOD, we present a demonstration case study. We have chosen a model featured as a tutorial in
the online documentation so that readers will be able to easily reproduce, run and edit it
(https://barneydobson.github.io/wsi/demo/scripts/oxford_demo/). The demonstration covers the area of Oxford,
UK, depicted in Figure 5 both as a map and schematic.

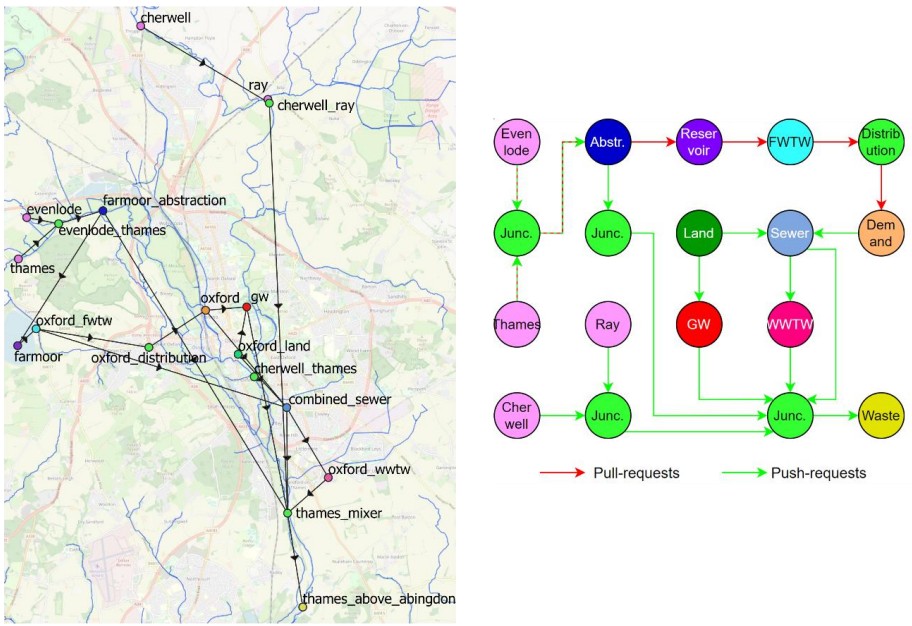


**Figure 5: (a, left) Map of the model nodes and arcs over the city of Oxford, map base layer © OpenStreetMap contributors, licensed under the Open Database License (ODbL), rivers data from Open Rivers © 2023 Ordnance Survey Limited (b, right) schematic demonstrating the node-arc connectivity of the model.**

The model includes four upstream catchments, a reservoir-based water supply system, and a combined sewage
system, all of which ultimately combine and drain into the River Thames. Because the model is primarily
demonstrative, the parameters given in the model are estimated based on local knowledge. However, high-
resolution (weekly sampling) water quality data are available in the area (Bowes et al. 2018) thus enabling accurate
boundary conditions at upstream catchments and water quality representations in the model. In Figure 6 we plot
simulations against the weekly sampling, water quality indicators to plot were selected based on those that had
data for all locations for the simulation duration.



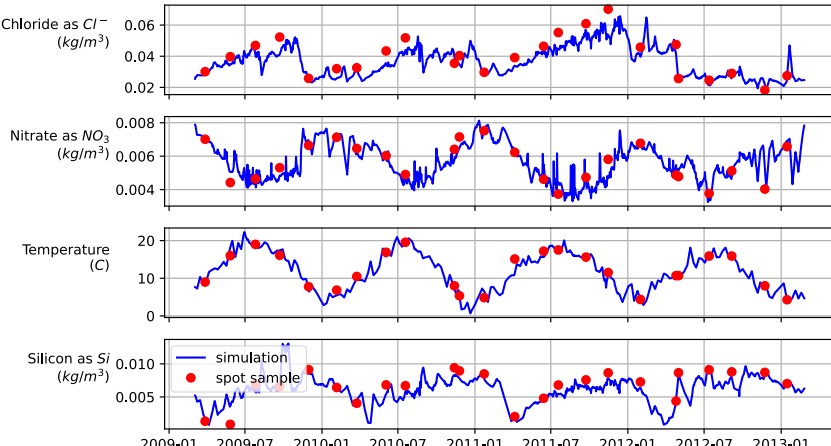


**Figure 6: Demonstration of simulated levels of chemical water quality indicators against spot samples.**
In addition to WSIMOD's capabilities as a general-purpose simulator of hydrological and water quality, it is also
a valuable tool for management and interventions. The software's design prioritizes customization, making it easy
to incorporate specific operational preferences which are not typical to capture due to the limited modelling scope
of most water simulators. For example, in the online customisation guides
(https://barneydobson.github.io/wsi/how-to/), we show how the behaviour of nodes and arcs can be altered to
accommodate changing abstraction licencing and environmental flow requirements for the Oxford case study,
with results shown in Figure 7. Liu et al., (2023b), demonstrate how a user can implement highly sophisticated
water quality management strategies in WSIMOD based on pollution load allocation.

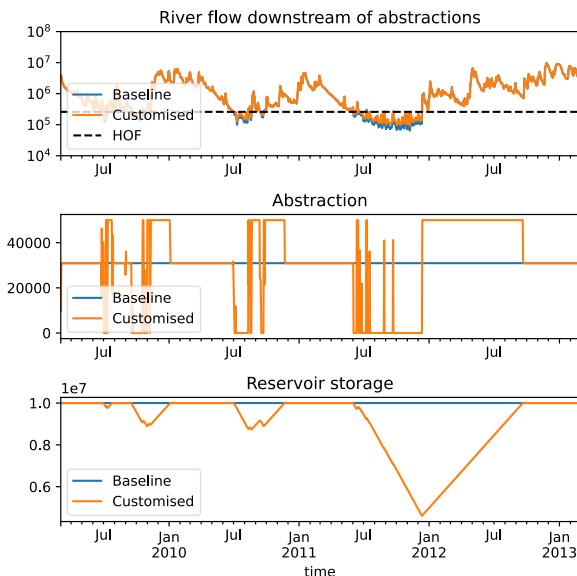


**Figure 7: Example of implementing a 'hands-off flow' (HOF), which abstractions cannot draw water below, for the**
**reservoir abstraction node in the Oxford case study model. The 'customised' simulation are with HOF implemented,**
**and 'baseline' simulation without.**



## 4 Discussion

### 4.1 Why are integrated water systems models necessary?

We defined four key goals for integrated models of water systems. In this section we will discuss how WSIMOD helps to meet these goals.

#### 4.1.1 Integration to understand emergent behaviour in water systems

Emergent behaviours in human and environmental systems arise from the interactions between multiple components and are difficult to predict and understand without addressing the complexity that occurs from this interconnectedness (Liu et al. 2007). The human-altered water cycle covers a diverse set of components and so any attempt to understand the fundamental (physical and operational) drivers behind emergent behaviours must acknowledge the interactions and feedbacks between humans and hydrological processes (Wada et al. 2017). WSIMOD represents many components within water cycle needed to capture these interactions to reveal and quantify fundamental processes, for example, that in-river water quality during wetter periods is driven by agricultural processes, and during drier periods by urban processes (Liu et al. 2022). Without integrated modelling, we risk oversimplifying the system and omitting feedbacks that could have significant implications for water management decisions (Dobson and Mijic 2020).

Understanding and reducing uncertainty behind behaviour in hydrological systems requires intercomparison of hydrological process representations and thus a flexible modelling framework (Knoben et al. 2019). We suggest the same is true for the wider water cycle, and so anticipate that the flexible approach to integrated modelling in WSIMOD is well suited for these purposes, for example, by comparing different assumptions around water consuming behaviour (Dobson et al. 2021). Furthermore, customisability enables accommodating unconventional water systems that may stress the assumptions of underlying process representations. This approach can help identify any weaknesses or gaps in the model and refine our understanding of the both the process in question and the behaviour of the wider system. For example, the poor simulations of hydraulic structures identified in Dobson et al. (2022) are a result of reduced accuracy in capturing water head, in turn caused by the discrete modelling scheme inherent to a non-tightly coupled integrated framework. Identifying this weakness has implications beyond simulating hydraulic structures and provides guidance for future work.

Finally, research that contributes to knowledge must be reproducible, and if it is not then it can hardly be considered science (Hutton et al. 2016). To ensure that WSIMOD applications are reproducible the software is open source with a permissive licence, and it is provided with significant documentation and worked examples to ensure that it is used as intended. This documentation transparently lists the assumptions made for each model component, both in the source code of that component and in a self-contained library page (Dobson et al. 2023c). Furthermore, the ability to save the Model class in a self-contained and human readable file enables publishing WSIMOD applications in an easy to reproduce way.

#### 4.1.2 Integration to expand model boundaries

The WSIMOD modelling framework offers a versatile approach to representing various processes that are commonly treated as boundary conditions in water cycle models. One example is the agricultural-hydrological



Land node, which utilizes the CatchWat model to estimate the pollution and flow in rivers upstream of the model region (Liu et al. 2022, 2023a). Accounting for upstream pollution and dilution is crucial in accurately assessing the impact of urban processes on in-river conditions. By incorporating the Land node, WSIMOD provides a means to capture the typically neglected upstream agricultural system in urban wastewater studies, which is often viewed as a boundary. WSIMOD also facilitates the representation of boundary conditions within urban systems. For instance, it allows for the inclusion of abstractions, which are critical in understanding in-river impacts of wastewater effluent because of their influence on dilution during low flows (Dobson and Mijic 2020).

Besides achieving accurate simulations, we further propose that a modelled representation of boundary conditions is essential towards assessment of the system under future scenarios. For example, by treating the upstream river as a model rather than as a fixed boundary condition, the in-river impact of the wastewater system for future scenarios can be quantified, which would not be possible if the changes to upstream river behaviour under the future scenario were fixed (Mijic et al. 2022).

### 4.1.3 Integration to evaluate intervention impacts at whole-water system scale

The OOP deployed in WSIMOD enables flexibility to incorporate a variety of different physical and management interventions to the water cycle. As demonstrated in Liu et al., (2023a), the included WSIMOD nodes may be customised both in terms of their parameters and in terms of their physical processes to represent a range of Nature Based Solutions (NBS) such as flood plains, runoff attenuation features, regenerative farming, urban green space, and urban wetlands. Because all WSIMOD nodes can communicate with all other nodes, this is a reasonably straightforward exercise as new rules for interactions with other model components do not need to be redefined when additional components are added to the model.

Due to the flexibility and coverage provided by WSIMOD models, it is often straightforward to include decision-making, policy constraints, and operational rules (Dobson and Mijic 2020). The high-level control over orchestration provides an ideal place to represent these stakeholder actions that require information on a multitude of states across the wider system. For example, the amount available to be abstracted from the River Thames, UK, is based off both reservoir levels and river flow, meaning no specific node has access to all of the information required to determine abstraction amount (Environment Agency 1991). Rather than creating a complicated interaction between London reservoirs and the River Thames, a modeller may more simply inspect the state of these two systems in the orchestration to dynamically set the abstraction capacity, filling the role of an operator (Dobson and Mijic 2020). Thus, the nationally important question of modelling changes to abstraction policies may easily be implemented in WSIMOD, and the impacts of these, both on water supply and in-river water quality, may be quantified (Mijic et al. 2022).

The parsimonious methods selected to represent physical components in WSIMOD ensure that simulations are computationally efficient. Further to this, if computational speed is of critical interest, then we have demonstrated that varying timestep and spatial resolution is possible while still achieving accurate simulations (Dobson, Watson-Hill, et al. 2022). The result is an integrated water cycle model that can be used for purposes that are not typically computationally tractable. Dobson et al., (2022), demonstrate exploration of uncertainty in sewer model parameters (roughness and runoff coefficient) is possible with WSIMOD, this study is the first of its kind because pipe network models are typically too computationally expensive to perform the numerous simulations required for uncertainty analysis. Meanwhile, Liu et al., (2023a), demonstrate that regional portfolios of 5-year catchment-



scale nature-based solutions can be created by applying optimisation to a WSIMOD model spanning 32
catchments.

### 4.1.4    Integration to align simulations with systems level outcomes

A key benefit we have found in applications of WSIMOD is that, due to the breadth of systems that are represented,
performance metrics that are relevant to different stakeholders can usually be included in the model in a physically
based way. The most obvious example, which we have drawn on throughout this paper, is the ability to place in-
river impacts central to decision making. This enables better alignment with policy goals, such as the Water
Framework Directive chemical water quality classifications, which are defined based on in-river average pollutant
concentration (Environment Agency 2020b). However, a variety of other metrics can be conjunctively included
such as the reliability of water supply. For example, Dobson & Mijic (2020) examine both the water quality and
water supply benefits of a variety of water cycle interventions (leakage reduction, rainwater harvesting, etc.). Such
an approach is particularly beneficial to stakeholder engagement because non-water facing stakeholders can still
be represented and understand their interactions with the water cycle and system-wide goals. For example, Puchol-
Salort et al., (2022), demonstrate how developers can measure the impacts of new developments on flooding,
water quality and water supply. Furthermore, this study demonstrates how integrated modelling can enable them
to quantify how to ensure developments achieve a net-zero impact on these metrics, interestingly revealing that
retrofitting households outside of the development area is typically required to offset changes.

### 4.2    Data to support water systems integration

A key challenge to the application of an integrated approach to water systems modelling is the difficulty of setting
up models and associated availability of data. In WSIMOD, we provide parsimonious but physically based
representations to ensure parameterisation is possible with widely available data and we propose model evaluation
with observed in-river flow and water quality. The catalogue of documented nodes in WSIMOD (Dobson et al.
2023c) presents data requirements to help users understand how feasible it will be to apply the approach in their
study area. The data requirements will be entirely dependent on what nodes and at what resolution a model user
chooses to represent. However, we provide a broad overview of data requirements for a generic catchment scale
WSIMOD case study in Table 2, which may help to develop future automatic model setup at national and global
scale to facilitate applicability.
**Table 2: List of datasets typically required in catchment-scale WSIMOD applications with references for data sources**
**or further information.**

| Category | WSIMOD input | Availability | Further information (scale) |
|---|---|---|---|
| Climate | Evapotranspiration | Global datasets available | (Khan et al. 2018), (global) |
| | Temperature | Global datasets available | (Morice et al. 2021) (global) |
| | Precipitation | Global datasets available | (Sun et al. 2018) (global) |
| Rural | Catchment outlines | Global datasets available | (Lin et al. 2019a) (global) |
| | Catchment connectivity | Global datasets available | (Lin et al. 2019b) (global) |
| | Catchment hydrologic parameters | Global datasets available for some hydrological models | (Zhang and Schaap 2018) (global) |
| | Crop surfaces | Global datasets available | (Thenkabail et al. 2016) (global) |
| | Crop properties | Lookup tables available | (Allen et al. 1998) (-) |



| | | | |
|---|---|---|---|
| | Pollutants/nutrients | National datasets may be available with high uncertainty | (Liu et al. 2022) (UK) |
| Urban | Population | Global datasets available | (Leyk et al. 2019) (global) |
| | Garden area | National datasets may be available, otherwise rule of thumb may be acceptable | Office for National Statistics (2021) (UK) |
| | Wastewater treatment plants | European dataset available through Urban Wastewater Treatment Directive | European Commission (2016) (Europe) |
| | Foul catchments | National datasets may be available, otherwise contacting wastewater companies required | (Hoffmann et al. 2022) (UK) |
| Water use | Irrigation water use | Global datasets available | (Thenkabail et al. 2009) (global) |
| | Water resources system | Not typically available, contacting water supply companies required | - |
| Evaluation | Flow observations | National datasets typically available | (Fry 2010) (UK) |
| | Water quality observations (river and WWTW) | National datasets may be available | (Environment Agency 2020a) (England) |
| | Reservoir levels | Not typically available, contacting water supply companies required | - |

## 4.3 Future work and research direction

One of the main concerns that other modellers have expressed regarding a WSIMOD-like approach is the level of detail with which components are represented. While hydrologists and agricultural modellers are often comfortable with parsimony and aggregation in their catchment modelling, most other parts of the water cycle tend towards more complexity in their representations. This desire for complexity is likely due to the detailed application context in which different models have been developed. For instance, designing a new process in a wastewater treatment plant inevitably requires a highly detailed model (Hreiz, Latifi, and Roche 2015). However, questions are being raised in many fields with a tradition of complex modelling about the need for such complexity. Models of in-river phosphorus (Jackson-Blake et al. 2017), urban flooding (Li and Willems 2020), and sewer flow (Dobson, Watson-Hill, et al. 2022; Thrysøe, Arnbjerg-Nielsen, and Borup 2019) have shown that good results can be achieved with simpler approaches. While practical modellers typically question what level of complexity is necessary to answer their questions, scientific modellers examine the impacts of assumptions to build evidence around whether they are suitable and under which circumstances. WSIMOD prioritises integration of the whole-water cycle, which is enabled by reduced-complexity modelling of the system components. In the examples provided, we demonstrate that sacrificing complexity in terms of detail should be viewed as an opportunity to better accommodate and contextualise components in the wider water cycle, as well as highlighting the importance of interactions between components.

We see the WSIMOD platform as an ideal opportunity for the environmental modelling community to implement and compare (or benchmark) different modelling assumptions and examine water cycle impacts. We plan to continue developing the representation of different components and testing whether more complex representations can improve simulations. Our focus will be on representations of a wider range of NBS, treatment plants, and urban sewer network hydraulic structures. Furthermore, we believe that complementing WSIMOD with machine learning representations of components that are too complex to be captured in a physical way can be a promising approach, thus implementing a "surrogate" strategy (Razavi et al. 2022; Razavi, Tolson, and Burn 2012).



A key opportunity for improving the accessibility of WSIMOD will be in the development of a graphical user
interface (GUI). The current implementation as a Python package makes the software well suited to customisable
and flexible simulations for programmers, but inaccessible to a wide range of potential users. We see a variety of
different approaches towards greater interactivity and visualisation that are not mutually exclusive. A "virtual
decision room" approach may provide an ideal environment for non-technical stakeholders to explore simulation
results and to highlight integrated system-wide impacts (Schouten, van den Hooff, and Feldberg 2016).
Meanwhile, incorporation into GIS-based frameworks such as 3DNet (Todorović et al. 2019) or Google Earth
Engine (Gorelick et al. 2017) would enable more seamless incorporation of pre-processing and provide a suite of
streamlined tools to help users create, edit, and run WSIMOD models.

**5    Conclusion**

We have presented the theoretical underpinning of WSIMOD, which is an open-source software for simulating a
range of urban and rural processes and operations in the integrated water cycle. WSIMOD represents different
components of the water cycle as nodes that are connected by arcs. The nodes that we have discussed throughout
the paper are parsimonious implementations that are conducive towards easy parameterisation and setup. Arcs
convey interactions between nodes that fall under four key categories: pushes (a node has water to go somewhere),
pulls (a node needs water from somewhere), requests (interaction represents flux of water and/or pollutants), and
checks (interaction does not represent flux). This integration framework allows all nodes to communicate with
each other, thus facilitating a flexible method that can accommodate a wide variety of water systems. Because
this approach uses object-oriented programming, WSIMOD enables customisation to capture unconventional
behaviours and implementation of a wide variety of physical and management interventions.
In summary, our early case studies show WSIMOD to be a useful and versatile tool for water systems modelling.
We hope to have persuaded other modellers of the importance of an integrated approach and believe the design
philosophy behind WSIMOD can serve as a helpful starting point for understanding integration in their respective
contexts.

**6    Competing interests**

The contact author has declared that none of the authors has any competing interests.

**7    Acknowledgements**

This work was funded by the NERC CAMELLIA project (Community Water Management for a Liveable
London), grant NE/S003495/1. LL is funded by the President's PhD scholarships provided by the Imperial College
London. The views expressed in this paper are those of the authors alone, and not the organisations for which they
work. We are grateful to Liliane Manny and Liu Bo for their insightful comments on the manuscript that have
improved the paper.



## 8 Code availability

WSIMOD is provided open-source under the terms of the BSD-3-Clause license. The code can be accessed at https://github.com/barneydobson/wsi (last access: 2023-07-19), and documentation at https://barneydobson.github.io/wsi/ (last access: 2023-07-19), with further technical details in Appendix 1. The code has been tested up to Python 3.10 and requires minimal dependencies (see website).

## 9 Author contributions

BD and LL created and tested all model code and documentation. All authors were involved in theoretical development and writing.

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

## 11     Appendix 1 – technical details in WSIMOD

### 11.1     A1.1 Tank object to generalise water stores

A particularly important concept in developing new and understanding existing nodes is the Tank object. The concept of a water store is so common in water systems that a generic Tank object is provided. Tanks streamline a variety of uses for stores and have a range of 'children' that implement travel time (A1.2) and pollutant decay (A1.3). A node that represents a water store should be a subclass of the Storage class (see Figure 2), which itself is a generic node wrapper for the Tank object. The simplest case of a tank would be a water supply reservoir, demonstrated for WSIMOD at a lumped London scale in (Dobson and Mijic 2020). However, many nodes use stores but in an auxiliary fashion, for example WWTWs have temporary storage tanks while FWTWs have service reservoir tanks.

### 11.2     A1.2 Non-flux pollutant changes

By default, everything tracked in a VQIP follows mass balance. However, in the water cycle, many pollutants undergo transformations due to biological, physical, or chemical processes, and thus preservation of mass may be insufficient to simulate them. WSIMOD represents the nitrogen/phosphorus cycles in soil (see documentation of



Land nodes) and denitrification/mineralisation/production/macrophyte uptake in rivers (see documentation of
River nodes), based on the equations from (Lindström et al. 2010; Liu et al. 2022).
While the transformations that act on chemicals in soils and rivers are well studied in the literature, there are
difficulties in conceptualising bio-chemical that take place in groundwater and sewers, despite agreement that
these are chemically active (Almeida, Butler, and Friedler 1999). As a result, WSIMOD provides a generic two-
parameter method to implement temperature sensitive chemical decay, given by:

|  | $M_t = M_{t-1}(1 - cd^{T_t - T_{ref}})$ | (1) |
|---|---|---|

, where M is the mass of a chemical in each timestep, c is a parameter that determines non-temperature sensitive
decay, d is a parameter that determines temperature sensitive decay, T is the temperature with a reference
temperature ($T_{ref}$) assumed to be 20C. We do not intend that equation (1) can be a substitute for well-researched
and verified process representations, however in our experience using WSIMOD it is an easy and useful option
to improve water quality representations.
Wastewater and freshwater treatment processes are well-studied fields. However, simulation models of these
systems require detailed information describing the different treatment technologies and processes that are present
in a specific plant. While we plan to include these types of models in WSIMOD in the future, we have opted to
take a parsimonious approach to treatment modelling under the assumption that most users will not have detailed
information about the plants they model. This approach assumes that the plant performs a single operation, based
on equation (1), to transform influent, which is then split into three streams of effluent, liquor, and solids.
Depending on whether freshwater or wastewater treatment, these streams go to different places, see documentation
of WTW for further details.

### 11.3   A1.3 Travel time of water

Arcs are the key model element to implement travel time of water. Two arc subclasses that provide alternate
methods to implement travel time are provided in WSIMOD. The first, more simple approach, formulates the
travel time of the arc as a dictionary object where each key is the number of timesteps remaining; when water is
sent along the arc, it is combined with the any existing water for the key that matches the specified travel time.
These travel times are updated at the end of each timestep. This method is computationally efficient because the
number of operations each timestep is limited by the maximum number of timesteps the arc takes to traverse.
However, this approach cannot represent a dynamic flow capacity, as is the case in, for example, sewer networks,
where hydraulic head governs flow. Thus WSIMOD also contains a less computationally efficient arc to
accommodate this behaviour, described and demonstrated in (Dobson, Watson-Hill, et al. 2022). Arcs can also
implement pollutant changes associated with decay over this travel time, using equation (1).