# Peer review of "Modelling water quantity and quality for integrated water cycle management with the WSIMOD software"

_EGUsphere, 2023_

## Author Response (AR1)

Dear Editor,

We thank you and the reviewers for the time in reviewing this manuscript. We have found the review process extremely helpful in improving the paper. We believe we have addressed all changes suggested by the reviewers. The two most significant changes include better motivating integrated water system modelling in the introduction by using examples, as suggested by reviewer 1, and creating a stronger linkage between the methods and the demonstration, as suggested by reviewer 2.

Best,

Barney Dobson

**Reviewer 1**

I enjoyed reading this manuscript. I also applaud the authors for such a nice contribution to the modeling community. Given that GMD is highly interdisciplinary and has a broad audience, I encourage the authors to better define some terminologies upfront. A few specific, minor comments are listed below.

We thank the reviewer for their encouraging and useful comments. Lines refer to the new manuscript with track changes enabled.

1. Line 26-28. "water cycle" --> "terrestrial water cycle" since the components listed here do not include atmospheric or ocean components.

It seems different people have different definitions or understandings about the "components" in a water cycle. To help the readers better understand, a conceptual diagram might be useful here to illustrate the various components (particularly those represented in the model) and their physical and anthropogenic linkages.

Lastly, by "hydrological catchments" do the authors mean the surface areas where rain falls and runoff is generated and routed into rivers? If so, I'd suggest placing it at the very beginning of this list, since the terrestrial water cycles (at the catchment, regional, or global scales) begin with rainfall and runoff processes.

We agree that this terminology around water cycle was not sufficiently clear, thus we have used terrestrial water cycle in the abstract and start of introduction and stated that our use of 'water cycle' refers to the terrestrial water cycle at Line 57.

We agree that a conceptual diagram in the introduction could be useful, however, Figure 1 is currently at the start of section 2 and so we did not want to have a similar diagram in

the introduction. To ensure a reader begins the paper with a visual understanding, however, we have used a simplified version of Figure 1 as a graphical abstract.

We agree that is makes sense to form the list of captured elements of the water cycle broadly from upstream to downstream, now re-ordered at Line 29.

2. Line 28. What is the difference between "hydrological catchments" and "river catchments"? If they are the same, please just use "catchments" after the definition upfront.

We agree that 'river catchments' was confusing, we have changed to 'hydrological catchments' throughout. 'Catchments' alone would have risked confusion with foul/groundwater catchments which is also used.

3. Line 74, "abstracts". Please use another word if possible to avoid confusion. For example, "abstraction" could mean "rainfall infiltration and/o retention" to some hydrologists.

Agree, changed to pumps.

4 . Line 457-460. It would be better to move this example upfront in the introduction, perhaps along with another couple of classic ones. This way, the readers will have a better sense of what are "boundary conditions" and then the authors intend to achieve. I did not get this puzzle resolved until reading to this point.

We agree that the introduction was currently missing some more tangible examples to properly motivate the need for integrated modelling. Thus, we have expanded and rearranged the paragraphs on the topic of 'problems of water system integration' to include a variety of interesting examples that can only be answered with an integrated model (Line 44-57):

*"We briefly pose a variety of questions to further motivate these challenges, some of which are developed throughout the paper. Can changes in agriculture fertilising behaviour offset the increased sewage due population changes downstream (Liu, Dobson, and Mijic 2022)? Will water efficient appliances concentrate household sewage to the extent that river water quality is worsened (Mott MacDonald 2023)? Will wastewater reuse worsen low flow conditions in rivers downstream of the wastewater treatment plant where it is implemented (Mott MacDonald 2023)? Can upstream water supply river abstractions be strategically reduced on sewer overflow days to dilute the spill without compromising supply reliability (Dobson and Mijic 2020)? Will increasing prevalence of 'working from home' change where sewage is produced to have knock-on impacts on wastewater infrastructure and rivers (Dobson et al. 2021)? In a multi-polluter hydrological catchment, can combinations of pollution reduction achieve a target water quality at a downstream checkpoint (Liu, Dobson, and Mijic 2023b)? For each of these questions, there will be water systems where these questions are true, and cases where they are not. We term these problems of water systems integration, and it follows that understanding the terrestrial water cycle ('water cycle' hereafter) as a whole is needed to address them."*

**Reviewer 2**

The authors present an open-source Python tool to make integrated model of the water cycle, including water quantity and quality. They start with a review of the state of the art, then present the conceptual structure of the model, very briefly introduce a demonstration and then focus on discussing the need for the tool. Finally they present some conclusions.

The work that the authors present is interesting and, in my opinion, within the scope of GMD. Their tool being a library, may serve to others researchers to adapt it to their needs, either using the already developed classes, either by modifying the code. I think the paper may deserved to be published in GMD, however I have a major concern about the paper structure.

We thank the reviewer for their encouraging perspective and constructive comments, we believe that we have addressed these satisfactorily, as set out below. Lines refer to the new manuscript with track changes enabled.

Section 2 presents a conceptualization of the model, which is not evident to follow. Such a conceptualization is important, because it allows to extend the model in an easy way and to represent similar elements with a minimal amount of code. However, conceptualizations are difficult to grasp, so I would suggest that the authors combine section 3 (which currently is not very illustrative, probably because the link provided is not working for me -I get a 404 error-) with section 2 in such a way that they first explain a canonical case in water management, that anyone could understand, and from there, they present the conceptualization.

In this way, the reader would be able to put "some meat" in the concepts presented, making it easier for the reader to apply the WSIMOD software to their own problems. Without such a change, the presentation may not be clear enough to ensure an understanding.

Moreover, after having checked the paper where the source code was published, I am also missing some more technical details. They may be better suited to supplementary material, but I believe that an explanation of the technical details and equations behind the model elements would be interesting. Not only to have all that information in a citable location, but also to have it all concentrated in one place, so that when problems arise, the users of the library may try to troubleshoot problems with the equations looking at just one section of the paper. (This information may be somewhere in the documentation, but I have not been able to find it).

We first apologise to the reviewer that the links to the online documentation were not working, the website and GitHub are now hosted by Imperial College London rather than the lead author's personal account. All links have now been updated.

We agree that further development of the conceptualisation was needed in Section 2, which we have addressed in a variety of ways:

- The Oxford case study is now used in examples throughout the paper rather than just in Section 3, it was unnecessarily confusing to have an underdeveloped Luton example only to introduce a new location in the demonstration.

- We have substantially revised the introduction to Section 2, both providing the requested step-by-step conceptualisation of the generic WSIMOD water cycle, and by providing better linking to the more technical elements of this section. Due to the amount of changes, we do not reproduce them here but direct the reviewer to Lines 160-197 in the revised manuscript. We opted to do this rather than completely combine the sections (which the reviewer suggests) because keeping the methods separate is more typical and ensures that a reader has the complete information required to understand the demonstration before starting it.
- We have included a new paragraph in the introduction that provides more tangible integration examples before getting into detail, at Line 44-57. We believe this will help readers arrive to Section 2 with more 'meat' to the concepts that are explained in the methodology.

We agree that it is important for users to be able to understand the technical assumptions behind the model. As the software is constantly evolving, we do not believe that a supplement or appendix would be a suitable format for this. The documentation of the software (see subsections of https://imperialcollegelondon.github.io/wsi/reference/ and https://imperialcollegelondon.github.io/wsi/component-library/ for a summary) does contain such information. As we identify issues with the documentation, we will raise these via GitHub issues to keep track and fix them. As this information was insufficiently visible from the original manuscript, we have included it at,

Line 195, new text:

*"If readers wish to be involved with the development of WSIMOD, identify bugs, or require further clarification in terms of documentation, we recommend the viewing Contributing section of the main repository page (https://github.com/ImperialCollegeLondon/wsi, accessed 2024-04-08)."*

Line 221, revised text:

*"We summarise these components in Figure 2 and the Component Library section in the online documentation, for full details we recommend viewing the API reference and 'Key assumptions' section of Node subclasses in the online documentation, as well as various tutorials on their use (Dobson et al. 2023b)."*

\* Minor comments

+ Line 19/20: Improve use of verbs.
Corrected

+ Line 201: "our" seems to be out of place.
Corrected

+ Line 354: adjudged may not be the term the authors intended to use
Corrected